# Vitamin B12 induces memory of predation through vitellogenin provisioning

Shiela Pearl Quiobe ⓘ , Ata Kalirad, Raphaela Zurheide, Hanh Witte, Christian Rödelsperger ⓘ & Ralf J. Sommer ⓘ ✉

Nutritional effects on trait inheritance are widespread in animals including human famines, representing some of the most severe environmental influences on organismal phenotypes and can be transmitted over generations. However, the chemical nature of the stimuli inducing such memory remains elusive. The nematode *Pristionchus pacificus* exhibits mouth-form plasticity including predation and responds to a multigenerational *Novosphingobium* diet with the induction and transgenerational inheritance of the predatory morph. We show that bacteria-derived vitamin B12 is necessary and sufficient for transgenerational memory. *Novosphingobium* mutants defective in vitamin B12 production do not induce transgenerational inheritance, but vitamin B12 supplementation can rescue the memory phenotype. Different vitamin B12 concentrations are required for the original induction and subsequent transgenerational inheritance of the predatory morph. This inherited effect acts through increased multigenerational vitellogenin transcription suggesting elevated nutrient provisioning. Consistently, mutants in the vitellogenin receptor *Ppa-rme-2* are memory-defective indicating that a vitamin acts through maternal provisioning to progeny. Thus, vitamin B12 induces vitellogenin provisioning to control organismal physiology and behavior.

Dietary and nutritional effects can alter organismal phenotypes and influence trait inheritance over generations, so-called transgenerational epigenetic inheritance (TEI)[1,2]. A large body of evidence indicates that dietary factors can induce various morphological, physiological, and behavioral changes which are often transmitted for three generations or more. However, the chemical nature of the inducing agents and downstream events following the original dietary stimuli are largely unknown. One major obstacle in identifying inducing stimuli from the diet and the genetic machinery underlying TEI in the receiving organism is the lack of model systems with robust phenotypes to be analyzed under laboratory conditions. However, nematodes such as *Caenorhabditis elegans* and *Pristionchus pacificus* with their short generation time, isogenic propagation, and simple husbandry, have started to provide mechanistic insight[3–7].

For instance, *P. pacificus* exhibits a mouth-form dimorphism that is an example of developmental plasticity with morphological and behavioral implications and therefore, provides a robust system to analyze dietary effects and TEI[8–10]. During postembryonic development, genetically identical *P. pacificus* worms adopt either the narrow-mouthed "stenostomatous" (St) morph with a single dorsal tooth, or the wide-mouthed "eurystomatous" (Eu) form with a claw-like dorsal and an opposing sub-ventral tooth (Fig. 1a). Importantly, only the Eu morph allows omnivorous feeding, including predation on other nematodes (Fig. 1a). Mouth-form plasticity is characterized as a bistable developmental switch allowing the underlying gene regulatory network (GRN) to be elucidated in detail[10–13]. Additionally, multiple environmental cues influence mouth-form development, including changes in diet, establishing an easy study system for experimental manipulation[14].

A recent study introduced a long-term environmental induction experiment with distinct diets by propagating 110 *P. pacificus* isogenic lines for 101 generations with associated food-reversal experiments (Fig. 1b)[3]. In addition to the standard nematode laboratory food *Escherichia coli* OP50, this study used the *Pristionchus* environment-

Department for Integrative Evolutionary Biology, Max Planck Institute for Biology Tübingen, Tübingen, Germany. ✉e-mail: ralf.sommer@tuebingen.mpg.de

**a**

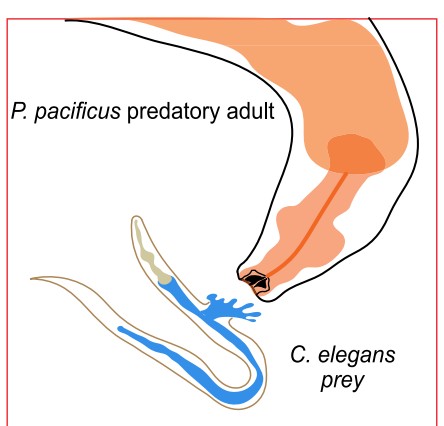

**b**

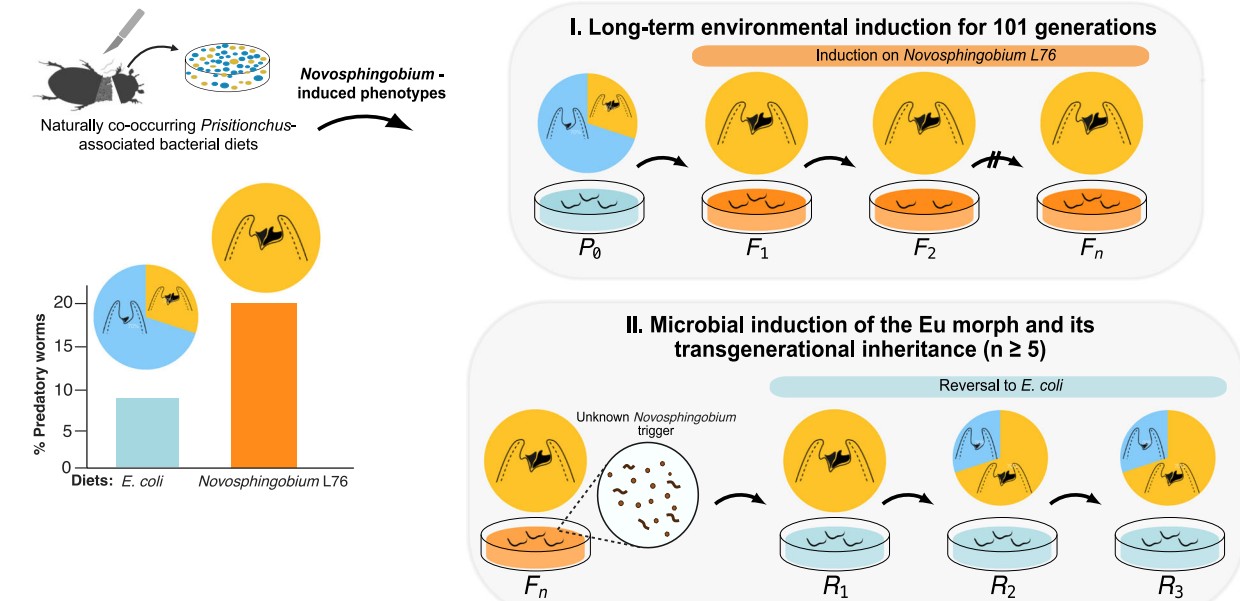

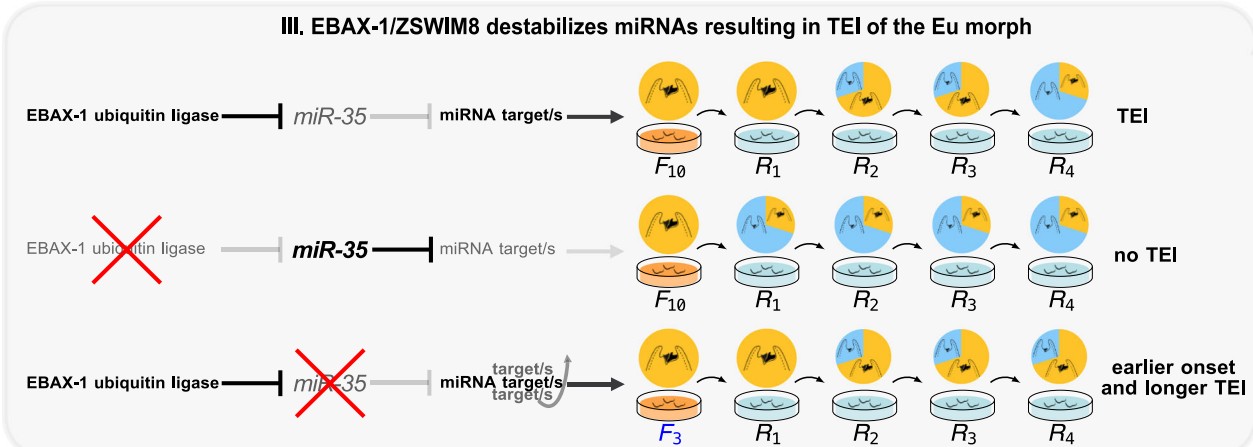

**Fig. 1 | A role for the ubiquitin ligase EBAX-1/ZSWIM8 in dietary-induced transgenerational inheritance of the predatory mouth form in *P. pacificus*.**
**a** Mouth dimorphism of *P. pacificus*. The eurystomatous (Eu) morph has a wide mouth with two teeth and feeds on bacteria and nematodes, whereas the stenostomatous (St) morph has a narrow mouth with a single tooth and feeds only on microbes. Right picture: Adult Eu *P. pacificus* devouring a *C. elegans* larval prey. **b** Graphical representation derived from Quiobe et al., 2025. Using long-term dietary induction and food-reversal experiments across 110 isogenic *P. pacificus* lines propagated for 101 generations demonstrated dietary induction and transgenerational inheritance of the predatory mouth form. Further studies identified EBAX-1/ZSWIM8–mediated destabilization of the expanded *miR-223Sa/miR-35* miRNA cluster as a key mechanism underlying this inheritance, consistent with a role for target-directed miRNA degradation.

derived bacterium *Novosphingobium* L76 that was originally shown to increase killing efficiency of *P. pacificus*[15]. The long-term environmental induction experiment study revealed diet-derived induction of the Eu mouth form during *Novosphingobium* L76 exposure and subsequent TEI of this predatory phenotype when worms were reverted to an *E. coli* diet (Fig. 1b). Interestingly, TEI of the Eu morph requires a minimal exposure of five generations on the inducing *Novosphingobium* L76 diet, which is phenomenologically different from most other examples of TEI that require only a short-term stimulus. Subsequent genetic studies identified a previously unknown mechanism in the regulation of TEI[3]. Forward genetic screens for mutants defective in transgenerational inheritance indicated an essential role of the ubiquitin ligase EBAX-1/ZSWIM8 and the destabilization of clustered microRNAs of the *miR-2235/miR-35* family for transgenerational memory. Specifically, *Ppa-ebax-1* mutants show no memory, whereas deletions of the *miR-35* cluster result in precocious and extended TEI of the predatory mouth form (Fig. 1b). While these studies start to provide molecular mechanisms underlying TEI, nothing is known about the factors from *Novosphingobium* that induce the predatory mouth form and its subsequent TEI.

In this work, we studied bacterial metabolites and found that vitamin B12 is both necessary and sufficient to induce the predatory mouth form and its subsequent memory in a concentration-dependent manner. This effect is inherited through increased vitellogenin transcription and consistent with this finding, mutations in the single vitellogenin germline uptake receptor *Ppa-rme-2* are memory-defective. Thus, vitamin B12 induces vitellogenin provisioning in a multigenerational manner to control organismal physiology and behavior.

## Results

### Vitamin B12 induces the predatory morph and transgenerational memory

Long-term environmental induction is performed by switching naïve *P. pacificus* RSC011 cultures that were always grown on the standard laboratory food source *E. coli* OP50 to *Novosphingobium* L76 (Fig. 1b). This *P. pacificus* strain from La Réunion Island is preferentially non-predatory (30%Eu:70%St) when grown on *E. coli*. The original long-term environmental induction experiment with 110 isogenic lines derived from the same ancestral RSC011 individual revealed dietary induction of the predatory mouth form (Fig. 1b). This dietary effect is (i) immediate (in generation 1 on *Novosphingobium*), (ii) complete (100% Eu progeny), (iii) systemic (in all 110 lines), and (iv) permanent (throughout the entire experiment of 101 generations). Following the reversal from *Novosphingobium* back to the standard *E. coli* OP50 diet, we observed TEI of the predatory mouth form (Fig. 1b). Importantly, such TEI is only seen after a minimal exposure to *Novosphingobium* for five generations, whereas a shorter exposure will not result in memory of the predatory morph.

We wanted to determine the molecular nature of the dietary stimulus inducing the predatory mouth form and its transgenerational inheritance in *P. pacificus* RSC011. The *Novosphingobium* L76 strain used in this experimental setup was originally isolated from a *Pristionchus*-associated environment and was shown to enhance killing efficiency of the *P. pacificus* wild type strain PS312, which is naturally 100% Eu[15]. This study also identified bacterial vitamin B12 production to cause enhanced predation, to accelerate worm development and increase brood size[15]. Therefore, we wanted to know if vitamin B12 would also play a role in dietary induction of the predatory mouth form and TEI in *P. pacificus* RSC011 animals. For that, we transferred single naïve J4 animals to vitamin B12-supplemented agar plates with *E. coli* OP50 as food source (Fig. 2a). It is important to note that *E. coli* OP50 does not produce vitamin B12 but rather obtains its vitamin B12 from the tryptone in the agar[15–17]. We originally used final

concentrations of 500, 1000, and 1500 nM of one of the two active forms of vitamin B12, methyl-Cobalamin (Me-Cbl), for supplementation, similar to previous studies in *P. pacificus*[15]. Note that these concentrations are substantially higher than those used in previous work in *C. elegans*[17]. Strikingly, we observed an immediate induction of the Eu mouth form after vitamin B12 supplementation at all three concentrations used (Fig. 2b–d and Supplementary Fig. 1b–d). However, vitamin B12 supplementation results in a strong but not a complete induction of the Eu mouth form. Specifically, in three independent biological replicates with a total of 60 assay plates, we found high but incomplete induction in the first generation after vitamin B12 supplementation ($0.878 \leq$ HDI $(\theta) \leq 0.994$, where $\theta$ is the inferred probability of developing the predatory mouth form). This effect was seen over a period of 5 and 10 generations and was observed in all lines (Fig. 2b–d and Supplementary Fig. 1b–d). Thus, vitamin B12 mimics *Novosphingobium*-based induction of the predatory mouth form in *P. pacificus*.

To determine if vitamin B12 would also induce TEI of the predatory mouth form, we established a mini-assay with a 5-generation or 10-generation exposure of worms to a vitamin B12-supplemented *E. coli* diet, followed by 5 generations on an un-supplemented diet (Fig. 2a–d and Supplementary Fig. 1a–d). Indeed, we observed TEI of the predatory mouth form (Fig. 2b–d and Supplementary Fig. 1b–d). Animals supplemented with 1000 and 1500 nM vitamin B12 showed TEI of the Eu mouth form similar to animals that were exposed to *Novosphingobium* for 5 or 10 generations (Fig. 2b, c and Supplementary Fig. 1b, c). In contrast, a supplementation with 500 nM of vitamin B12 only caused the induction of the predatory mouth form but no full TEI of the Eu morph (Fig. 2d). Specifically, the response during reversal only lasted for two generations, with all following generations being down to baseline Eu frequencies ($0.841 \leq$ HD1$(\theta_{R1}) \leq 0.987$, $0.52 \leq$ HD1$(\theta_{R2}) \leq 0.842$, and $0.305 \leq$ HD1$(\theta_{R3}) \leq 0.637$) (Fig. 2d and Supplementary Fig. 1d). Such a response is referred to as "intergenerational", but not transgenerational inheritance[4]. These results allow two major conclusions. First, vitamin B12 supplementation mimics diet-derived induction of the predatory mouth form and its subsequent transgenerational inheritance. Second, the effect of vitamin B12 is concentration-dependent with only 1500 and 1000 nM but not 500 nM of vitamin B12 causing TEI.

### Vitamin B12 induces a distinct transcriptomic response

Next, we wanted to know whether vitamin B12 supplementation can elicit similar effects on gene expression as those caused by the *Novosphingobium* diet. In principle, the *Novosphingobium* diet and vitamin B12 supplementation might induce similar transcriptomic responses or alternatively, have limited overlap in individual genes only. We examined the overlap of differential gene expression (DGE) found between vitamin B12 supplementation on *E. coli* OP50 with *Novosphingobium*-induced gene expression changes in RSC011 animals by using single worm transcriptomics of day 1 adult individuals (Fig. 2g). After three generations of exposure, *Novosphingobium*-upregulated genes were in parts also upregulated upon vitamin B12 supplementation (Fig. 2g and Supplementary Data 1). Specifically, of the 530 upregulated genes after a three-generation vitamin B12 supplementation, 103 (12.2% of all uniquely upregulated genes) were also upregulated on *Novosphingobium*. Notably, these shared genes correspond to 19.4% of the vitamin B12-induced and 24.6% of the *Novosphingobium*-induced transcriptional responses. Analysis of PFAM enrichment among these shared genes revealed that Lipase GDSL (PF00657.26) and Lectin_C (PF00059.25) were enriched in both conditions, suggesting that lipid metabolism and carbohydrate-binding/cell-surface signaling form a core transcriptional response. Beyond this overlap, vitamin B12-specific genes were uniquely enriched for CUB (PF00431.24), associated with extracellular protein interactions,

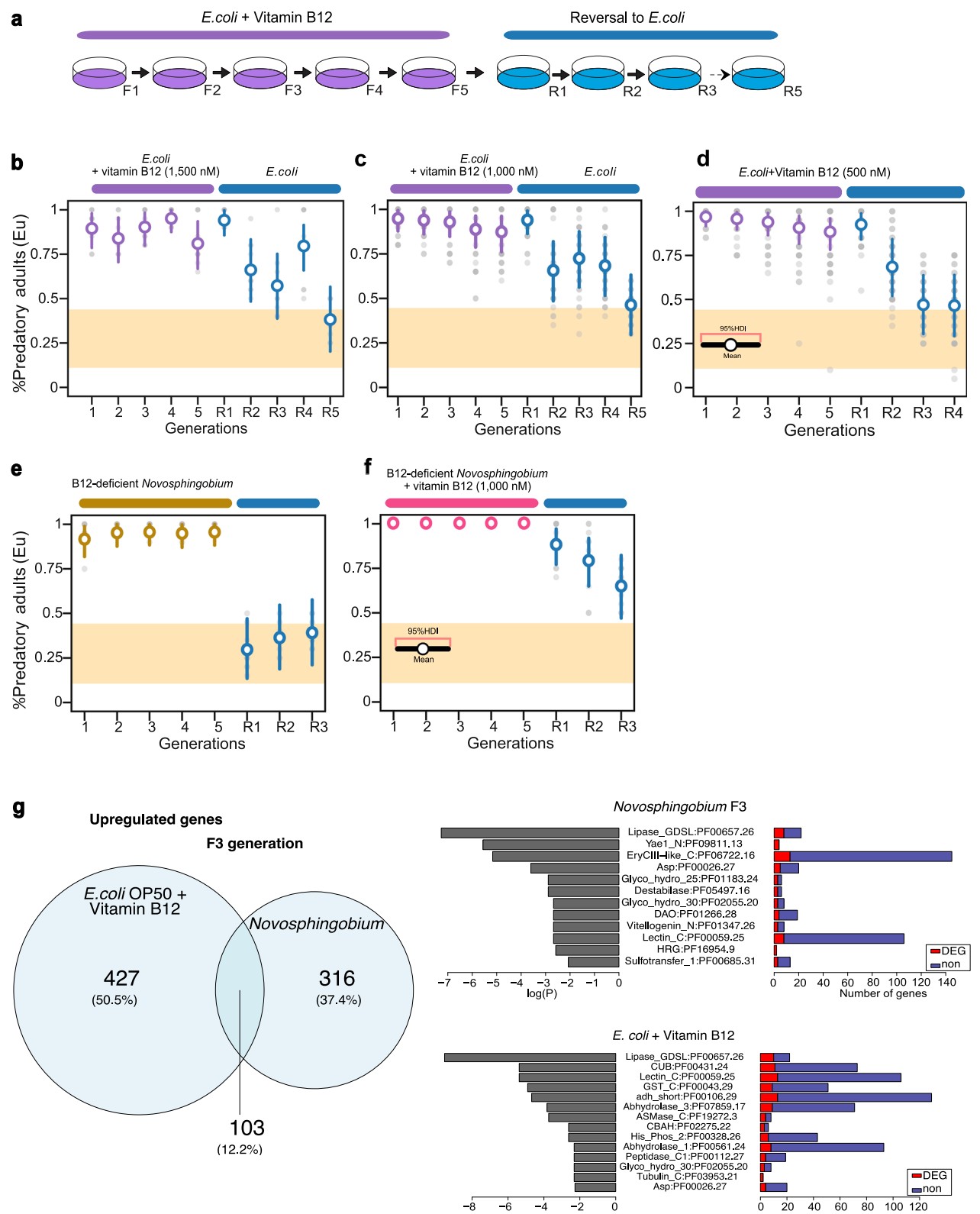

**g** Upregulated genes

F3 generation

whereas *Novosphingobium*-specific genes were enriched for Yae1_N (PF09811.13) and EryCIII-like_C (PF06722.16), reflecting ribosome-associated functions and secondary metabolite biosynthesis, respectively. These results highlight both conserved and condition-specific molecular programs activated in response to vitamin B12 supplementation and *Novosphingobium* exposure. (Fig. 2g and Supplementary Data 1). Thus, vitamin B12 supplementation to the *E.*

*coli* OP50 diet exhibits limited overlap with *Novosphingobium*-induced gene expression changes.

## Vitamin B12 is necessary and sufficient to induce memory

Previous studies that established a role of dietary vitamin B12 on life history traits in *P. pacificus* and *C. elegans* were based on screening Tn5-based bacterial mutant libraries[15,17]. We used a *Novosphingobium*

**Fig. 2 | Diet-derived vitamin B12 induces the predatory mouth form and its transgenerational inheritance. a** Schematic diagram for vitamin B12 supplementation showing five generations on vitamin B12-supplemented plates and subsequent exposure to non-supplemented *E. coli* plates. **b–d** Vitamin B12 supplementation mimics induction and subsequent transgenerational inheritance of the Eu mouth form. Mean probability of predatory mouth form for five generations on 1500, 1000, and 500 nM vitamin B12-supplemented *E. coli* and five generations on non-supplemented *E. coli*. **e** *Novosphingobium* vitamin B12-deficient mutant induces the predatory morph but not its transgenerational inheritance. Mean probability of the predatory mouth form on a *Novosphingobium* vitamin B12-deficient mutant and reversal to *E. coli*. **f** Rescue of transgenerational inheritance of the predatory mouth form by supplementing exogenous vitamin B12 to *Novosphingobium* vitamin B12-deficient mutant. Mean probability of predatory mouth form for 5 generations on vitamin B12-supplemented *Novosphingobium* mutants

and subsequent reversal to *E. coli*. Final mouth-form frequencies are the mean of at least 10 biological replicates (*n* = 20 animals per plate). Points represent the mean probability of developing the Eu morph, with error bars reflecting the 95% HDI from the Bayesian model. The yellow area indicates the RSC011 baseline response on *E. coli*, averaged over 101 generations from a previous study[3]. **g** Differential gene expression and enrichment analyses through single-worm transcriptomics (SWT). Exposure to *Novosphingobium* and vitamin B12-supplemented *E. coli* revealed overlapping differentially-expressed transcripts relative to worm cultures grown on a standard *E. coli* diet. The pathways with most significant enrichment (FDR-corrected *p* value < 0.01) for three generations on *Novosphingobium* and B12-supplemented *E. coli* are shown. SWT was performed using five independent biological replicates of young adult individuals. Naïve animals on *E. coli* were used as reference for differential expression analysis. See also Supplementary Fig. 1. Source data are provided as a Source Data file.

vitamin B12-deficient mutant to study the effect of *Novosphingobium*-derived vitamin B12 on mouth-form plasticity and TEI of the predatory morph. Using the 5-generation mini-assay described above, we found that a *Novosphingobium* vitamin B12-deficient diet would still induce the Eu mouth form (Fig. 2e). However, in contrast to a wild-type *Novosphingobium* diet, the induction is not complete and plateaus ~ 0.95% Eu $(0.818 \leq HDI\ (\theta_{F1}) \leq 0.98$ and $0.883 \leq HDI(\theta_{F5}) \leq 0.999)$. These results indicate that additional factor(s) besides vitamin B12 are able to induce the predatory mouth form. Thus, the complete (100%) induction of the Eu morph of *P. pacificus* RSC011 on *Novosphingobium* is due to multiple factors.

In contrary, we found that the *Novosphingobium* vitamin B12-deficient diet would not cause TEI of the predatory mouth form. When we reverted worm lines that were exposed to the *Novosphingobium* vitamin B12-deficient diet for 5 generations back to *E. coli*, we observed an immediate return to the base line response level $(0.135 \leq HDI(\theta_{R1}) \leq 0.47)$ (Fig. 2e). Similarly, when we reverted RSC011 lines that had been exposed to this diet for 10 generations, we also did not observe TEI of the predatory mouth form (Supplementary Fig. 1e). In contrast, vitamin B12 supplementation for 5 or 10 generations to the *Novosphingobium* vitamin B12-deficient diet restored the normal TEI response after reversal to *E. coli* OP50 $(0.772 \leq HDI(\theta_{R1}) \leq 0.971, 0.65 \leq HDI(\theta_{R2}) \leq 0.919$, and $0.47 \leq HDI(\theta_{R3}) \leq 0.832)$ (Fig. 2f and Supplementary Fig. 1f). These results indicate that vitamin B12 is both necessary and sufficient to induce the TEI of the predatory mouth form.

### Vitamin B12-induced TEI of the predatory morph requires *metr-1*

In animals, including humans, vitamin B12 acts as a cofactor of two enzymes, methionine-synthase in the cytoplasm and methylmalonyl coenzyme A (CoA) mutase in mitochondria (Fig. 3a)[18]. In *P. pacificus* as in *C. elegans*, the methionine-synthase is encoded by *Ppa-metr-1*. To manipulate the methylmalonyl coenzyme A (CoA) mutase pathway, we targeted the upstream factor *Ppa-mce-1*, that is one-to-one orthologous to *Cel-mce-1*. We targeted *Ppa-metr-1* and *Ppa-mce-1* in *P. pacificus* RSC011 by CRISPR to study a potential role in dietary induction and TEI of the predatory mouth form (Fig. 3b). We obtained two frameshift alleles of *Ppa-metr-1* and *Ppa-mce-1* each, and tested all of them in our standard assay (Fig. 3b). First, we studied the response of *Ppa-metr-1* mutants to an *E. coli* diet supplemented with 500, 1000 and 1500 nM vitamin B12. *Ppa-metr-1* mutant animals failed to respond to vitamin B12 indicating that the role of vitamin B12 in inducing the Eu mouth form requires *Ppa-metr-1* (Fig. 3c). Second, we grew *Ppa-metr-1* mutants on *Novosphingobium* and observed a mean induction of $\bar{\theta}$ = 0.96 throughout the exposure of 5–10 generations (Fig. 3d and Supplementary Fig. 2a, b). This result is likely due to the other *Novosphingobium* factor(s) that can induce the Eu mouth form as indicated in the experiments using the *Novosphingobium* vitamin B12-deficient diet (Fig. 2e and Supplementary Fig. 2e). Importantly, when we

reverted *Ppa-metr-1* mutants from *Novosphingobium* back to *E. coli*, we observed no TEI of the predatory mouth form (Fig. 3d and Supplementary Fig. 2a, b). These findings indicate that vitamin B12 acts as a cofactor of methionine-synthase in the transgenerational inheritance of the Eu mouth form. Third, we tested *Ppa-mce-1* mutants and observed an induction of the Eu mouth form on *Novosphingobium* similar to wild-type animals (Fig. 3e and Supplementary Fig. 2d, e). After reversal, *Ppa-mce-1* mutant animals responded similar to wild type controls, indicating that *Ppa*-MCE-1 is not involved in the TEI of the predatory mouth form (Fig. 3e and Supplementary Fig. 2d, e). Note however, that the response of *Ppa-mce-1* mutants to vitamin B12-supplemented *E. coli* diet only resulted in a partial Eu induction $(0.256 \leq HDI\ (\theta) \leq 0.616$ for the highest vitamin B12 concentration) (Supplementary Fig. 2c). Together, our mutant analysis identifies a requirement of *Ppa-metr-1* for the vitamin B12-mediated induction of the predatory mouth form and its subsequent TEI.

### The Eu morph is already induced by 0.1 nM vitamin B12 supplementation

Given that the induction of the Eu morph and the transgenerational memory of the predatory mouth form require different concentrations of vitamin B12, we next wanted to know the minimal amount of vitamin B12 necessary to induce the Eu morph. While our original experiments used a final concentration between 500 and 1500 nM Me-Cbl (Fig. 2), we now reduced the concentration of vitamin B12 for several orders of magnitude (Fig. 4). Supplementation experiments using 250 nM and 125 nM vitamin B12, resulted in an immediate Eu mouth-form induction, but reversal experiments after 5 or 10 generations resulted in an even shorter memory than supplementation with 500 nM vitamin B12. Specifically, the response seen after 250, 125, and 50 nM vitamin B12 lasted for only one generation (Fig. 4a–c and Supplementary Fig. 3a, b). When we reduced the concentration of vitamin B12 even further and supplemented with 10, 5, 2, and 0.1 nM vitamin B12, all four of these concentrations still resulted in a strong induction of the Eu mouth-form (Supplementary Fig. 3c–f). In contrast, worms had lost their memory already in the F5R2 generation (for 50 nM, $0.276 \leq HDI(\theta_{R2}) \leq 0.655$, for 10 nM, $0.238 \leq HDI(\theta_{R2}) \leq 0.612$, and for 5 nM, $0.27 \leq HDI(\theta_{R2}) \leq 0.645)$ (Supplementary Fig. 3c–f). Thus, the reduction of vitamin B12 concentration in supplementation experiments results in the loss of TEI of the predatory mouth form, while the induction of the Eu morph remains high. When we reduced the vitamin B12 concentration even further, we found no induction of the predatory mouth form at a concentration of 0.01 nM vitamin B12 (Supplementary Fig. 3g). Instead, worms treated with this amount of vitamin B12 stayed at the baseline level of Eu mouth-form, similar to un-supplemented growth conditions. Therefore, TEI of the predatory mouth form requires a substantially higher vitamin B12 concentration than the initial induction of the Eu mouth form.

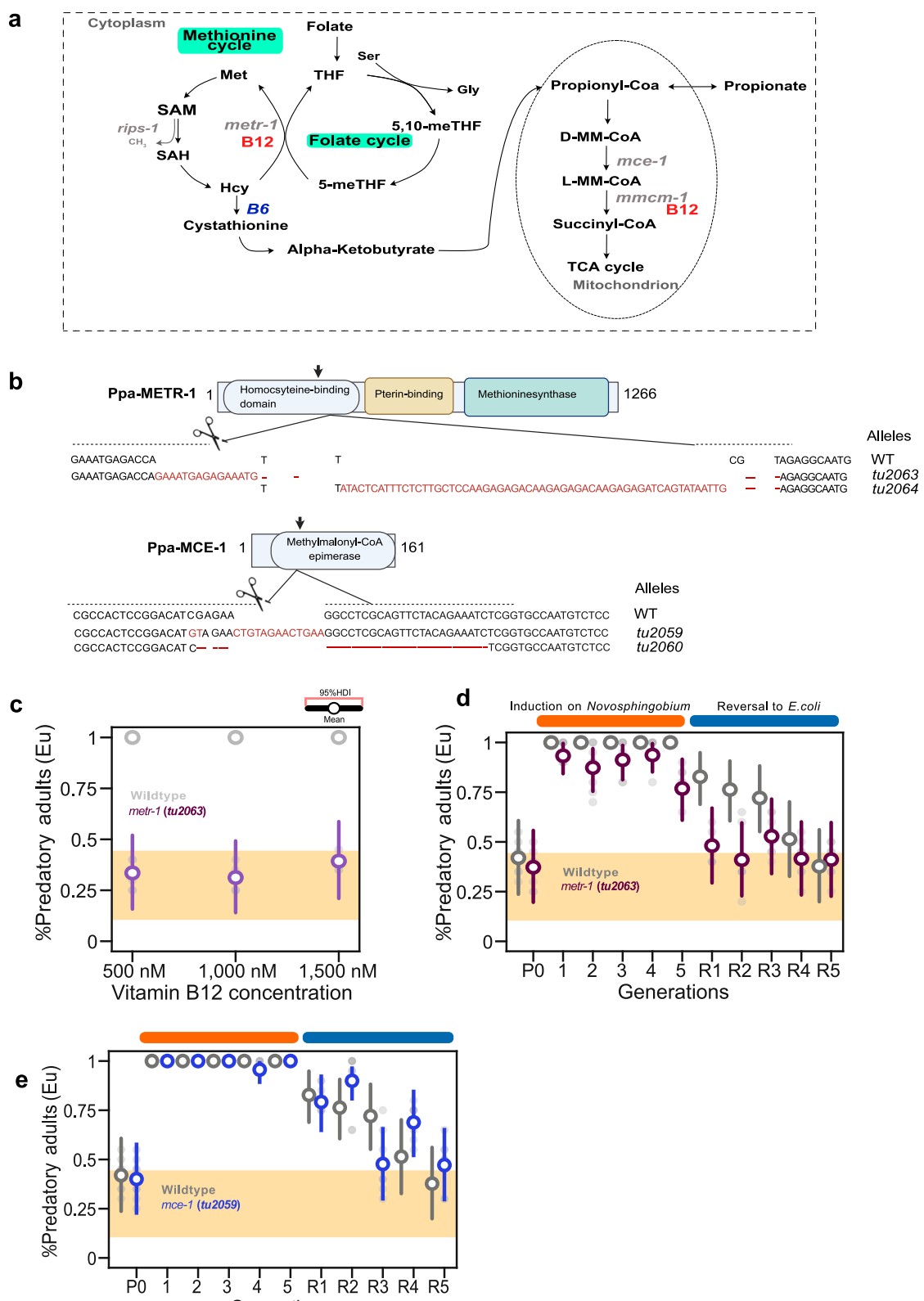

## Memory of the predatory morph requires methionine but not folate

Next, we used supplementation experiments with two major metabolites in the one-carbon cycle (Fig. 3a). First, we used 10, 50, and 100 μM of folate (vitamin B9) for supplementation of standard *E. coli* plates using similar range of concentrations as a previous study in *C. elegans*[19]. However, we found no credible induction of the predatory

mouth form (e.g., for 100 μM, $0.384 \leq HDI(\theta_{F2}) \leq 0.755$, Fig. 4d–f). Note that this might be due to the micromolar doses of folate that had to be used in these experiments due to solubility. In contrast, methionine supplementation resulted in the induction of the predatory mouth form and subsequent TEI; however, with different dynamics than what is observed after vitamin B12 supplementation. Specifically, supplementation with 5, 10, and 20 mM methionine resulted in the induction

**Fig. 3 | Vitamin B12 functions as a cofactor for methionine synthase *metr-1* in the transgenerational inheritance of the Eu morph. a** Requirement for vitamin B12 as cofactor for two enzymes in the cytosol and mitochondria. In the cytoplasm, it acts as cofactor of methionine synthase (*metr-1*) as part of the one-carbon cycle. In the mitochondria, vitamin B12 is a cofactor of methylmalonyl coenzyme A mutase. Both genes (*metr-1* and *mce-1*) encoding the metabolic enzymes in which vitamin B12 acts as a cofactor are shown. Red, vitamin B12; Blue, vitamin B6. **b** CRISPR/Cas9-induced mutations in *Ppa*-METR-1 and *Ppa*-MCE-1 with target locations indicated in respective protein domains (sgRNA, arrow). Molecular lesions of isolated mutations via CRISPR/Cas9 are also shown. **c, d** Requirement of *Ppa-metr-1* in transgenerational inheritance of vitamin B12-induced Eu mouth form. **c** Mean probability of predatory mouth-form in *Ppa-metr-1* mutant animals on vitamin B12-supplemented *E. coli* compared to wild type. **d** Mean probability of the predatory mouth form in *Ppa-metr-1* mutant animals on a *Novosphingobium* diet and reversal to *E. coli* compared to the wild type response. **e** Mean probability of the predatory mouth form in *Ppa-mce-1* mutant animals exposed to *Novosphingobium* and reversal to *E. coli* compared to wild type response. Final mouth-form frequencies are the mean of at least 10 biological replicates (*n* = 20 animals per plate). Points represent the mean probability of developing the Eu morph, with error bars reflecting the 95% HDI from the Bayesian model. The yellow area indicates the RSC011 baseline response on *E. coli*, averaged over 101 generations from a previous study[3]. See also Supplementary Fig. 2 and Supplementary Table 1. Source data are provided as a Source Data file.

of the Eu mouth form after several generations of exposure (Fig. 4g–i). Note that this supplementation does not result in 100% Eu animals similar to supplementation with lower concentrations of vitamin B12 (Fig. 4). However, reversal after five generations of exposure to methionine indicates a TEI of the predatory mouth form that lasts for three generations (Fig. 4g–i). Given that methionine significantly alters osmolarity, we controlled for this effect using sorbitol[20]. However, sorbitol supplementation did not induce the Eu morph, indicating that the observed response is specific to methionine rather than osmotic changes (Supplementary Fig. 3h). Overall, these results suggest that vitamin B12 might exert its function through methionine in the one-carbon cycle.

Given that vitamin B12 and methionine supplementations can both cause TEI of the predatory morph, we performed single worm transcriptomic analysis of methionine-supplemented worms of day 1 adult individuals. The overlap of upregulated genes between methionine supplementation and *Novosphingobium* exposure was limited (6.8%), whereas vitamin B12 supplementation showed a larger overlap with *Novosphingobium* (12.2%), indicating that vitamin B12 recapitulates a larger subset of the bacterial-induced transcriptional program (Fig. 4j). Notably, the overlap between significantly upregulated genes after vitamin B12 and methionine supplementation represents the largest fraction of 19% (Fig. 4j). Together, these patterns suggest that vitamin B12 accounts for a larger fraction of the *Novosphingobium*-induced transcriptional responses.

## Vitamin B12 causes elevated nutrient provisioning

It is important to determine what type of changes act downstream of the original stimulus to cause the inherited effect. This is of particular importance given the different vitamin B12 concentrations necessary to induce the predatory mouth form and its TEI. One possibility would be that vitamin B12 is directly transmitted from the mother to the offspring. Alternatively, TEI might be a consequence of factors acting downstream of vitamin B12. To test for the former possibility, we generated mutants in *mrp-5*, a gene that was shown in *C. elegans* to be responsible for the transport of vitamin B12 from mother to the offspring[21]. Indeed, *P. pacificus* has a 1:1 ortholog of *mrp-5* and we were able to create two mutations in *Ppa-mrp-5* that result in frameshifts (Fig. 5a). However, *Ppa-mrp-5* mutants show a normal TEI of the predatory mouth form (Fig. 5b, c). These findings suggest that TEI is unlikely to be driven by direct vitamin B12 transmission and instead depends on downstream factors or alternative transport mechanisms.

In principle, a multigenerational vitamin B12-rich diet could induce a plethora of transcriptional changes in the worm. Therefore, we extended our gene expression analysis to cultures after reversal and compared transcriptional profiles of single naïve, F3, and F5 generation worms on *Novosphingobium* with F3R2, F4R2, F5R2, and F10R2 generations after reversal to *E. coli* (Fig. 5d). We used FnR2 rather than FnR1 generations to rule out any effect of the kanamycin treatments in the FnR1 generation. Note that we could not use single worm transcriptomic analyses for this large set of conditions and instead, performed these experiments by harvesting cultures with preferentially

adult individuals. We found a consistently strong increase in the expression of vitellogenin genes, encoding the nematode yolk proteins, after *Novosphingobium* exposure with multiple vitellogenin genes remaining at higher expression levels in comparison to naïve animals (FDR-corrected *p* value < 0.01) (Fig. 5e). Note that vitellogenin genes were also upregulated in our single worm transcriptomic dataset described above (Fig. 2g), but several of them were not significant. As vitellogenins are transmitted from mothers to offspring, they represent potential candidates for a function in TEI of the predatory mouth form.

*P. pacificus* shows a more complex vitellogenin composition when compared with *C. elegans*. While the *C. elegans* N2 genome contains six vitellogenin genes, *P. pacificus* RSC011 has 9 genes, all of which are most closely related to *Cel-vit-6* (Fig. 5f, g). We therefore named these genes *Ppa-vit-6-A* to *Ppa-vit-6-I* based on their location along the chromosome (Fig. 5f, g). While more similar to one another in sequence than the *Cel-vit* genes, the divergence between the *Ppa-vit* genes still allows the distinction of individual genes in gene expression analysis (Fig. 5e). We found that 8 of the 9 vitellogenin genes are upregulated upon *Novosphingobium* exposure, four of which show an increased expression of at least eightfold in the F10 generation (Fig. 5e and Supplementary Data 2). A similar increase in expression was seen in all other tested generations on *Novosphingobium*. After reversal to *E. coli* the majority of vitellogenin genes were less expressed than on *Novosphingobium*; however, expression remained substantially higher than in naïve animals that had never been exposed to *Novosphingobium* (Fig. 5e). Thus, exposure to a multigenerational *Novosphingobium* diet and subsequent reversal to *E. coli* cause elevated vitellogenin transcription resulting in increased nutritional provisioning.

## Receptor-mediated endocytosis (*rme-2*) mutants have no memory

The correlation between elevated vitellogenin expression and the formation of the predatory mouth-form during and after *Novosphingobium* exposure suggests a role of nutritional provisioning in transgenerational memory. To provide mechanistic evidence for this hypothesis, we investigated the receptor involved in vitellogenin uptake into the germline. In nematodes, vitellogenin is produced in the intestine, and the low-density lipoprotein receptor (LDLR) encoded by *rme-2* (*r*eceptor-*m*ediated *e*ndocytosis protein) was shown to represent the single protein involved in vitellogenin uptake into the germline[22,23]. In *C. elegans*, *rme-2* mutant embryos contain no detectable yolk and the brood size of homozygous mutant animals is strongly reduced[22]. In *P. pacificus*, there is a 1:1 ortholog *Ppa*-RME-2 with a 55% amino acid sequence similarity (Fig. 6a). Strikingly, we had isolated three mutant alleles of *Ppa-rme-2* in our EMS mutagenesis screen for predatory mouth-form Transgenerational-inheritance-defective (Tid) mutants reported previously (Fig. 6a and Supplementary Table 2)[3]. This finding would be consistent with a role of vitellogenin in memory formation.

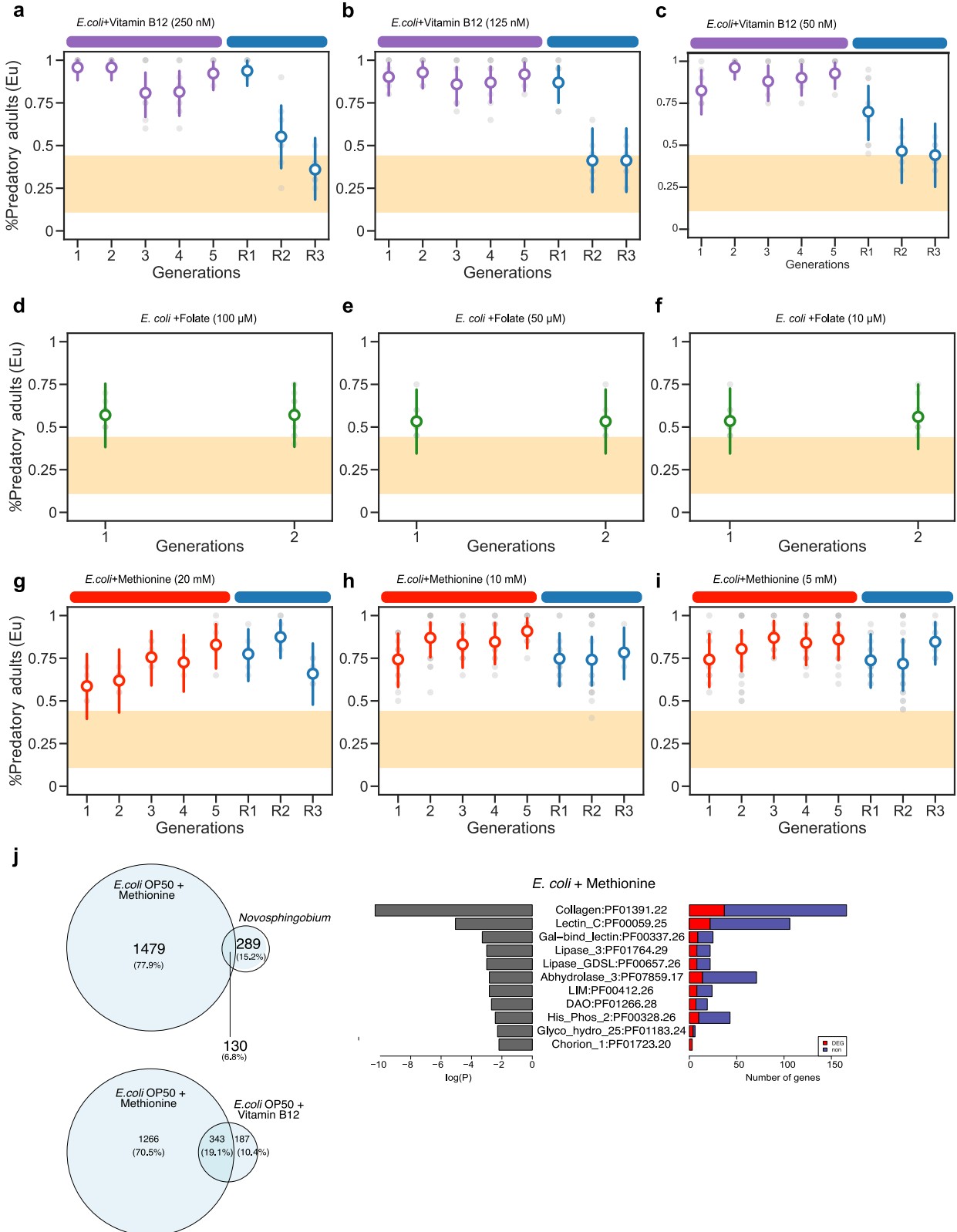

To rule out the possibility that EMS-induced mutations in genes other than *Ppa-rme-2* have resulted in the observed Tid phenotype, we generated an additional knock-out mutation by CRISPR engineering. The *Ppa-rme-2(tu2155)* allele has a 16 bp insertion that results in a premature stop codon (Fig. 6a). When we tested the *Ppa-rme-2(tu2155)* allele in a 5-generation *Novosphingobium* assay, we observed an incomplete induction of the Eu morph for all five generations of

*Novosphingobium* exposures ($0.64 \le \text{HDI}(\theta_{F1\text{-}tu2155}) \le 0.928$, $0.664 \le \text{HDI}(\theta_{F2}) \le 0.94$, and $0.628 \le \text{HDI}(\theta_{F5}) \le 0.921$) (Fig. 6b). This result indicates that in the absence of RME-2 and vitellogenin uptake into the germline, the predatory mouth-form can still be induced, which is likely due to other *Novosphingobium*-derived factors as previously indicated (Fig. 2c). In contrast, after the reversal to *E. coli*, *Ppa-rme-2(tu2155)* did not show transgenerational memory of the predatory

**Fig. 4 | Memory of the induced predatory mouth form after supplementation with different concentrations of vitamin B12, methionine, and folate. a–c** Concentration-dependent effect of vitamin B12 on memory transmission. **a** Mean probability of predatory mouth-form after a 5-generation exposure to 250 nM vitamin B12 shows Eu memory for two generations. Further reduction of vitamin B12 concentration to **b** 125 nM, **c** 50 nM, shows a single generation of Eu memory on un-supplemented *E. coli* diets. nM, nanomolar. **d–f** Mean probability of the predatory mouth form on folate-supplemented *E. coli* plates with **d** 100 μM, **e** 50 μM, and **f** 10 μM working concentration of folate. **g–i** Mean probability of the predatory mouth form on methionine-supplemented *E. coli* plates with **g** 20 mM, **h** 10 mM, and **i** 5 mM working concentration of methionine. mM millimolar. μM micromolar. Final mouth-form frequencies are the mean of at least 10 biological replicates

(*n* = 20 animals per plate). Points represent the mean probability of developing the Eu morph, with error bars reflecting the 95% HDI from the Bayesian model. The yellow area indicates the RSC011 baseline response on *E. coli*, averaged over 101 generations from a previous study[3]. **j** Differential gene expression and enrichment analyses from single-worm transcriptomics (SWT). Exposure to methionine-supplemented *E. coli* showed upregulated overlapping transcripts with *Novosphingobium* and vitamin B12-supplemented *E. coli*. Differentially-induced genes are identified relative to worm cultures grown on a standard *E. coli* diet. The pathways with most significant enrichment (FDR-corrected *p* value < 0.01) for three generations on methionine-supplemented *E. coli* are shown. SWT was performed using five independent biological replicates of young adult individuals. See also Supplementary Fig. 3. Source data are provided as a Source Data file.

morph       $(0.12 \leq \text{HDI}(\theta_{F5\text{-}tu2155}) \leq 0.431,$       $0.147 \leq \text{HDI}(\theta_{F5R2}) \leq 0.484)$ (Fig. 6b). Thus, vitellogenin is involved in the formation of transgenerational memory of the predatory mouth-form in *P. pacificus* indicating a role for nutrient provisioning in this process.

## Discussion

This work demonstrates that a food-derived vitamin can cause transgenerational memory of a morphological trait and its associated behavior over multiple generations. The identification of vitamin B12 as inducing stimulus for the TEI of the predatory mouth form in *P. pacificus* after multigenerational exposure to *Novosphingobium* indicates the significance of bacterial metabolites for animal development, growth and behavior and provides links to vitamin B12 deficiency in humans. Vitamin B12 (cobalamin) is the largest and most complex vitamin[18,24]. Exclusively synthesized by bacteria and some archaea, vitamin B12 is essential for animals, including humans. However, not all bacteria produce vitamin B12 and the standard *C. elegans*/*P. pacificus* food source, *E. coli* OP50, is a non-vitamin B12 producer. Therefore, under standard laboratory growth conditions, vitamin B12 is one of the growth-limiting factors for *E. coli* bacteria and worms, and both obtain vitamin B12 exclusively from the tryptone in the agar plates[16]. Consistently, previous work indicated that vitamin B12 supplementation or a vitamin B12-rich *Comamonas* or *Novosphingobium* diet affect several *C. elegans* life history traits and killing efficiency in *P. pacificus*[15,17,25,26].

Bacterial and animal requirement for vitamin B12 is low. For example, for *E. coli*, it was estimated that only 20 cobalamin molecules per cell are sufficient to support growth[27]. In humans, the total body content of vitamin B12 is estimated to be 1–5 mg with a daily requirement of 2–3 μg[28,29]. In *C. elegans*, studies by Bito and co-worker have shown that culture of worms under strict vitamin B12-deficient conditions will result in a gradual decrease of the vitamin B12 content over five generations[16]. In the fifth generation, *C. elegans* had only 4% of vitamin B12 relative to worms growing under standard conditions and started to show a loss of fertility and reduced lifespan at these reduced concentrations[16]. These findings are consistent with three of our observations; (i) the multigenerational exposure to a *Novosphingobium* diet being necessary to induce TEI of the predatory mouth form, (ii) the capability of vitamin B12 supplementation to induce TEI of this phenotype, and most importantly (iii) the concentration-dependent effect of vitamin B12 supplementation. It is important to note that most studies in nematodes used unphysiological amounts of vitamin B12 supplementation and in *P. pacificus*, we have to use even higher concentrations than in *C. elegans* to see TEI of the predatory morph. However, vitamin B12 is not directly taken up by the worms from the agar and instead is ingested through the *E. coli* diet. Given that the direct measurement of vitamin B12 still represents a major challenge, the exact amount of vitamin B12 in *C. elegans* or *P. pacificus* after a *Comamonas* or *Novosphingobium* diet is currently unknown, a limitation that is common to all studies on vitamin B12[30].

The experiments with *Ppa-mrp-5* mutants show a normal TEI, suggesting that it is not the direct transmission of vitamin B12 but a downstream factor causing the observed memory. However, the transgenerational effect is slightly decreased compared to wild type and one possibility that cannot be completely ruled out is that multiple transporters in addition to *Ppa-mrp-5* are involved in vitamin B12 transmission. Nonetheless, *Ppa-mrp-5* is one-to-one orthologous to *Cel-mrp-5*; thus, there are no additional copies of this transporter in the *P. pacificus* genome.

In *P. pacificus*, the inherited effect of vitamin B12 acts through vitellogenin. Vitellogenins are a family of yolk proteins representing the most abundant proteins in oviparous animals. Previous work in *C. elegans* had indicated a role of vitellogenins for post-embryonic development and fertility[21]. Increased vitellogenin provisioning is associated with several post-embryonic phenotypic alterations, i.e., advanced maternal age. Specifically, young *C. elegans* mothers provide less embryonic vitellogenin, resulting in early-hatched larvae being smaller and slower to reach adulthood. In contrast, older mothers provide more vitellogenin, resulting in larger larvae that are also more starvation resistant, indicating an intergenerational signal that mediates the influence of parental physiology on progeny[23,31]. In addition, it is important to note that vitellogenins are part of a lipoprotein complex which also carries lipids. These lipids are also dependent on RME-2 and recent studies in *C. elegans* have indeed indicated that they are involved in intergenerational inheritance[32,33].

The work described in this study has to be seen in the context of the ecology of *P. pacificus* nematodes. The majority of *Pristionchus* species is found in association with scarab beetles, where nematodes attach as dauer larvae to adult beetles[34]. After the death of the insect in the soil, *Pristionchus* and other nematodes compete for the limited food source on the beetle carcass. Through the ability to consume bacteria, fungi, and other nematodes, *Pristionchus* has an advantage in this heterogeneous environment, a phenomenon referred to as intraguild predation[35]. Therefore, the observed transgenerational memory might represent an advantage for subsequent generations. Interestingly, we have previously shown that *P. pacificus* can selectively feed on vitamin B12-producing bacteria in a semi-natural beetle carcass habitat[36]. In conclusion, our work extends recent findings indicating that nutrient provisioning affects organismal physiology and behavior involving transgenerational inheritance and requires epigenetic factors. This multigenerational effect highlights the significance of exceptions to the Weissmann barrier and suggests the transmission of vitellogenin and potentially other lipids from somatic tissues into the germline to have important consequences for development and evolution.

## Methods
### Experimental model and subject details
The ancestral *P. pacificus* RSC011 isolate used in this study was frozen within the first 5–10 generations after original isolation from Coteau Kerveguen on La Réunion to minimize domestication and thereby,

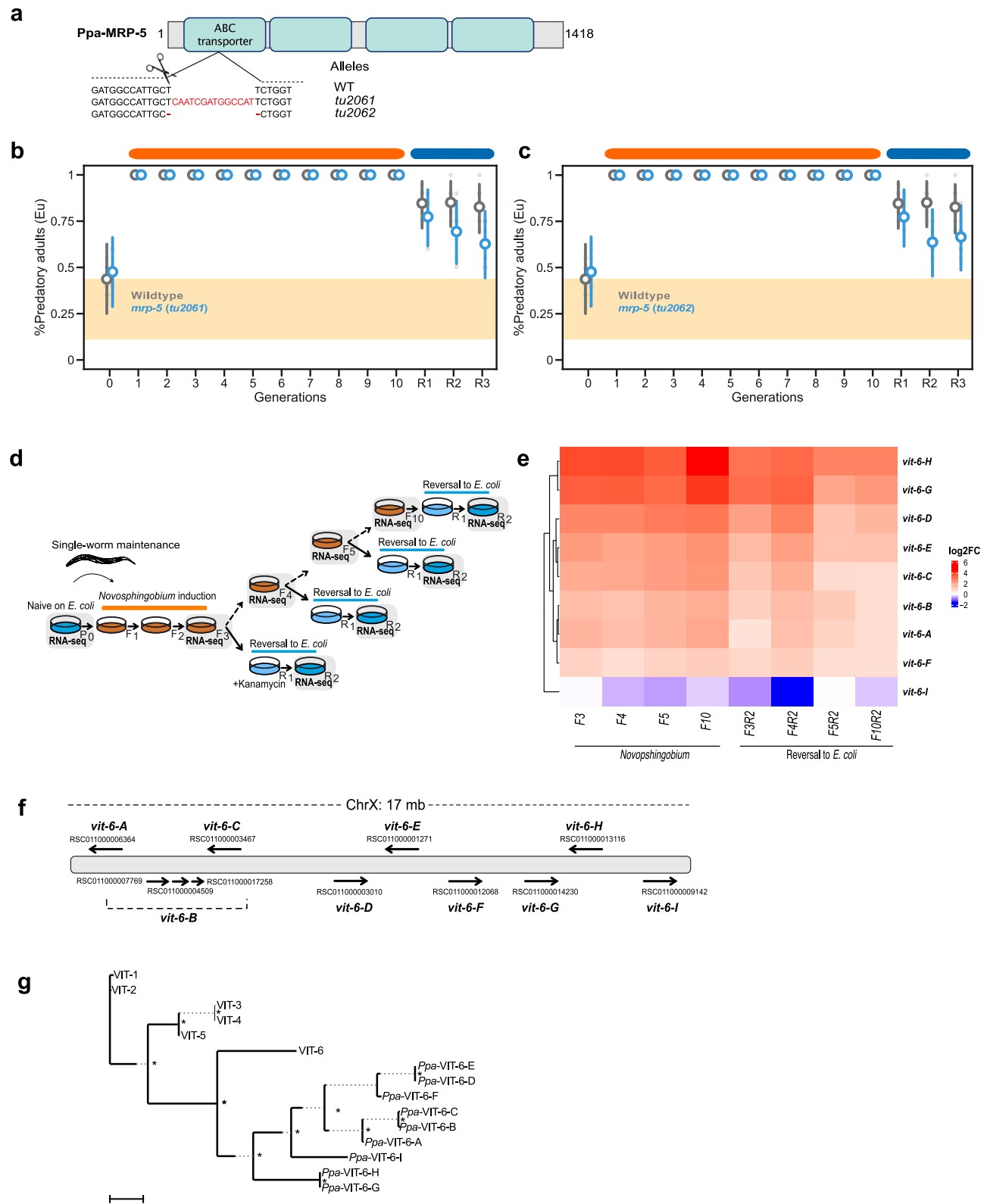

facilitate the investigation of diet-induced plasticity and TEI. Nematodes were grown under standard nematode growth conditions on NGM plates seeded with either *E. coli* OP50 or *Novosphingobium* L76 and maintained at 20 °C. The RSC011 stock was not subjected to bleaching, starvation, extreme temperature fluctuations as they are known to influence mouth-form ratios. Nematode mutant strains described in this study (*Ppa-metr-1*, *Ppa-mce-1*, and *Ppa-mrp-5*, and *Ppa-*

*rme-2*) were generated for this study and are available upon request from the lead author.

### Bacterial strains conditions

All bacterial strains and mutants were grown overnight in LB (Lysogeny broth) supplemented with 50 µg/ml kanamycin, where required to initiate food-reversal. Bacteria were grown at 30 °C or 37 °C,

**Fig. 5 | *Ppa*-MRP-5 is dispensable in TEI of the predatory morph and vitellogenin gene cluster organization. a** CRISPR/Cas9-induced mutations in *Ppa*-MRP-5 with target locations indicated in respective protein domains (sgRNA, arrow). Molecular lesions of isolated mutations via CRISPR/Cas9 are also shown. **b**, **c** Mean probability of predatory mouth-form in *Ppa-mrp-5* mutant animals (*tu2061* and *tu2062*) on *Novosphingobium* for 10 generations compared to wild type. Final mouth-form frequencies are the mean of at least 10 biological replicates (*n* = 20 animals per plate). Points represent the mean probability of developing the Eu morph, with error bars reflecting the 95% HDI from the Bayesian model. The yellow area indicates the RSC011 baseline response on *E. coli*, averaged over 101 generations from a previous study[3]. **d** Graphical representation of the experiment. Grey-highlighted generations indicate populations sent for RNA sequencing. Mixed worm cultures from three agar plates were used to assay independent generational datapoints during *Novosphingobium* induction and reversal experiments (F3R2,

F4R2, F5R2, F10R2). **e** Vitellogenin expression of all 9 genes under different wild-type conditions: Log$_2$FoldChange (log$_2$FC) values are shown for worms on a *Novosphingobium* diet (F3, F4, F5, F10) and reversal to *E. coli* (F3R2, F4R2, F5R2, F10R2). Naïve animals on *E. coli* (P0) were used as reference to generate log$_2$FC values. **f** *P. pacificus* RSC011 has 9 vitellogenin genes all located on the X chromosome, distributed over a 17 Mb range. These genes are named *Ppa-vit-6-A* to *Ppa-vit-6-I* based on their location along the chromosome. Note that there are three transcripts that encodes for *Ppa-vit-6-B*. **g** Phylogeny of the *P. pacificus* and *C. elegans* vitellogenin genes. In *C. elegans*, six vitellogenin genes show higher sequence divergence with the *Cel-vit-6* gene being the most diverse gene copy. Note that it is *Cel-vit-6* that is most similar to the individual genes in *P. pacificus* and also other nematodes. Nodes with bootstrap values of ≥90 are labelled with asterisks (*). Source data are provided as a Source Data file.

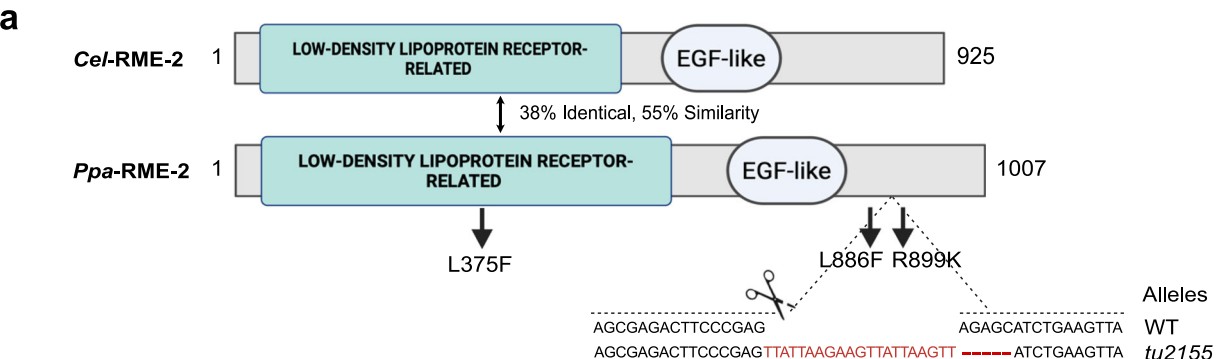

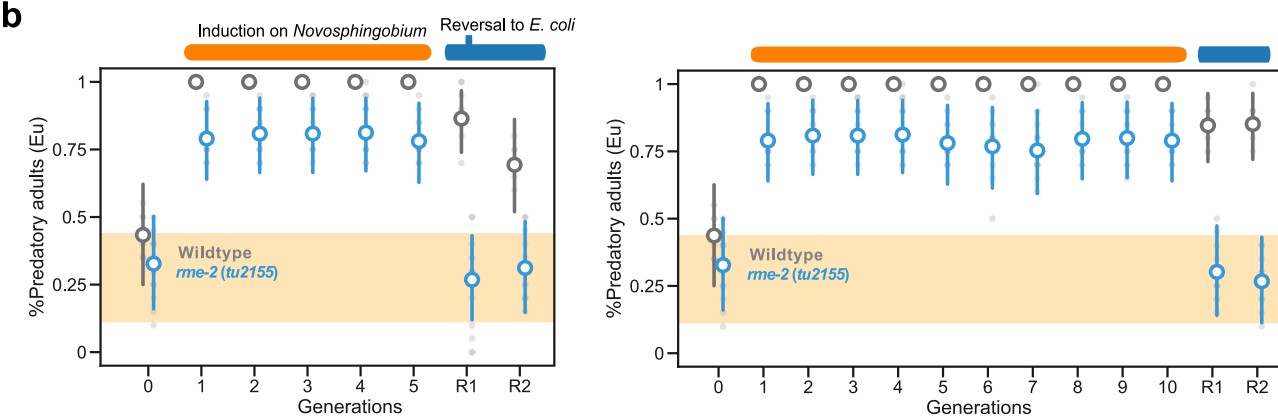

**Fig. 6 | The low-density lipoprotein receptor-related protein RME-2 is required for transgenerational inheritance of the predatory morph. a** Domain architecture of *C. elegans* and *P. pacificus* RME-2, sequence similarity, and position of molecular lesions in three *Ppa-rme-2* mutants are highlighted. Three alleles isolated in a forward genetic EMS mutagenesis screen carry point mutations resulting in amino acid changes as indicated. See also Supplementary Table 2 for the isolated alleles from the genetic screen. We designed a sgRNA in the C-terminal region where two of the original EMS alleles carry their mutation. The frameshift allele

*tu2215* was used for further analysis. **b** Mean probability of the predatory mouth form of *Ppa-rme-2(tu2215)* after a *Novosphingobium* exposure of 5 (left) and 10 (right) generations. Final mouth-form frequencies are the mean of at least 10 biological replicates (*n* = 20 animals per plate). Points represent the mean probability of developing the Eu morph, with error bars reflecting the 95% HDI from the Bayesian model. The yellow area indicates the RSC011 baseline response on *E. coli*, averaged over 101 generations from a previous study[3]. Source data are provided as a Source Data file.

depending on the species and 6 cm nematode growth medium (NGM) plates were seeded with 300 µl bacterial overnight cultures and were incubated for 2 days. The *Novosphingobium* L76 strain and the vitamin B12-deficient mutant were isolated and generated by our group and are also available upon request.

### Nematode culture and dietary reversal experiments
Overnight cultures of *E. coli* OP50 and *Novosphingobium* L76 were spread to NGM plates and incubated at room temperature for 2 days. The *P. pacificus* RSC011 strain is preferentially St (20–40% Eu) on a

standard *E. coli* OP50 diet. *Novosphingobium* exposure is performed by picking single J4 larvae onto a *Novosphingobium* L76 diet. J4 worms were initially left for at least an hour on an initial *Novosphingobium* diet to reduce traces of *E. coli* OP50 before transferring to final F1 *Novosphingobium* plates. Reversal experiments to an *E. coli* OP50 diet were performed after exposing of RSC011 worms for various numbers of generations, i.e., 1, 3, 5, 10, or 15 generations on *Novosphingobium* L76. Worms were initially transferred from *Novosphingobium* L76 to NGM plates supplemented with 50 µg/ml of kanamycin for one generation, then back to *E. coli* OP50-seeded NGM plates for subsequent

generations. Exposure to kanamycin for a single generation to eliminate traces of *Novosphingobium* did not affect mouth-form response[3].

## Mouth-form phenotyping

Mouth-from phenotyping was performed using Zeiss Discovery V.20 stereo microscope (X150 magnification) by observing the nematode buccal cavity based on mouth-form identities[9]. Final mouth-form frequencies are the mean of at least 10 replicates, each assaying 20 animals from a single plate.

## Statistics and reproducibility

To estimate the probability of developing the probability of developing the Eu mouth form in *P. pacificus* (**θ**), we constructed a hierarchical Bayesian model as previously described[3]

$$
\begin{aligned}
y_i &\sim \text{Bernoulli}(\theta) \\
\theta &\sim \text{Beta}(\omega(\kappa - 2) + 1, (1 - \omega)(\kappa - 2) + 1) \\
\omega &\sim \text{Beta}(\alpha, \beta) \\
\kappa &\sim \text{Gamma}(3, 1)
\end{aligned}
\tag{1}
$$

where **θ** is calculated for each replicate and hyperparameters ω and κ link the biological replicates for a given generation under a given experimental condition[37]. Each experimental condition included independent biological replicates consisting of 5–10 plates, with up to 60 plates assayed depending on the condition, and 20 animals per plate. The hierarchical model accounts for replicate-level variation across plates. The model was fitted to the laboratory measurements using PyMC[38] in Python 3.11 with Numpy 1.25.2[39]. The mean highest density interval (HDI) for **θ** for a group of observations was used to visualize the inferred probability of developing the Eu mouth form. We ensured that the estimated **θ** values were stable using common convergence diagnostics with effective sample size (ESS) $\geq 10000$[40].

All experiments were performed with independent biological replicates, and results were consistent across replicates. No statistical method was used to predetermine sample size. No data were excluded from the analyses. The experiments were not randomized because experimental groups were defined by predefined treatment conditions and generations. The investigators were not blinded to allocation during experiments and outcome assessment because the experimental conditions and mouth-form phenotypes were readily identifiable during scoring.

## CRISPR/Cas9-induced mutations

Mutations in candidate genes were generated using CRISPR/Cas9 methodologies[41]. Specifically, gene-specific crRNAs and universal trans-activating CRISPR RNA were obtained from Integrated DNA Technologies. Equal volumes of (5 µl) of each 100 µM stock were mixed and denatured at 95 °C for 5 min, then allowed to anneal at room temperature. Cas9 endonuclease (New England Biolab) was added to the hybridized product and incubated at room temperature for 5 min. TE buffer was used to adjust to final concentrations of 18.1 µM for the sgRNA and 2.5 µM for Cas9. The resulting mixture was microinjected into the germline of 40–50 *P. pacificus* RSC011 young adults. Eggs from injected P0s were collected up to 16 h post-injection. After hatching and two days of growth, the F1 progeny was segregated onto individual plates until they had fully-developed and laid eggs sufficiently. F1 genotypes were screened using Sanger sequencing and mutations were identified prior to isolation of homozygous mutants. Details of sgRNAs and primers used in this study are provided in the supplementary table. Evidence for null alleles was based on frameshift mutations causing premature stop codons in protein-coding sequences.

## Metabolite supplementation assays

Methylcobalamin (Vitamin B12 CAS No. 13422–55–4), L-methionine (CAS No. 63–68–3), and folate (CAS No. 59-30-3) were purchased from Sigma and dissolved in water at the highest possible soluble concentrations to prepare stock solution. To quantify real phenotypic effects and control for altered NGM osmolarity, D-sorbitol (CAS No.) was added[20] at a final working concentration of 5, 10, and 20 mM. All stocks were prepared fresh before use in each experiment. Metabolite solutions were mixed with NGM agar at the required concentration just before pouring to 6 cm plates. Plates were allowed to dry at room temperature for two days and then spotted with *E. coli* OP50. We first tested different concentrations of vitamin B12 and found the strongest and most reliable transgenerational epigenetic memory effect with a concentration of 1000 and 1500 nM, which is most likely unphysiological. Similarly, in *C. elegans*, dose-dependent effects have been seen for vitamin $B_{12}$[17]. Different concentrations for methionine and folate supplementation were based on previous studies[15,19].

## RNA sequencing and data analysis

We used single worm transcriptomics (SWT) of naïve and RSC011 worms reared on *Novosphingobium* for three generations using five independent replicates[42]. *E. coli* OP50 supplemented with vitamin B12 and methionine were also processed with the SWT protocol. In contrast, we used mixed cultures for assaying independent generational datapoints during *Novosphingobium* induction and reversal experiments in F3R2, F4R2, F5R2, and F10R2 generations. For that, worms were harvested from three NGM plates. In general, worms were pelleted for RNA extraction and total RNA was isolated using Direct-Zol RNA Mini prep kit (Zymo Research) following the manufacturer's instructions. The RNA-seq library preparation and sequencing were performed by the company Novogene. Raw reads were aligned to the *P. pacificus* RSC011 reference genome using Hisat2[43] (version 2.1.0), and read counts were quantified with featureCounts[44] based on the RSC011 gene annotations. For the analysis of vitellogenin expression, we combined raw counts of the three transcripts (RSC011000007769, RSC011000004509, and RSC011000017258) into a single *vit-6-B* count. Differential gene expression analysis was performed in R (version 4.0.3) using DESeq2 (version 1.18.1)[45]. For single-worm RNA-seq samples, genes with padj < 0.05 were considered differentially expressed to account for higher inter-individual variability. For mixed-worm RNA-seq samples, a more stringent cutoff of padj < 0.01 was applied, reflecting the lower variability and increased confidence in expression differences. Tests for overrepresentation of protein domains in sets of differentially expressed genes were performed using a Fisher's exact test in R. We used the FDR method as implemented in the p.adjust function in R to correct for multiple testing and only retained results with adjusted *P*-value < 0.01.

## Phylogenetic analysis

One-to-one orthologous genes between *P. pacificus* and *C. elegans* was identified from BLASTP searches and were validated by phylogenetic analysis. Proteins were aligned using Clustal Omega[46] and the output FASTA file was uploaded to the IQ-TREE tool[47,48]. Analysis was performed under default settings using the auto substitution model, and 1000 bootstrap alignments were calculated with ultrafast setting. FigTree was used to visualized the resulting phylogenetic tree (http://tree.bio.ed.ac.uk/software/figtree/).

## Reporting summary

Further information on research design is available in the Nature Portfolio Reporting Summary linked to this article.

# Data availability

Sequencing data that was generated for the current study has been submitted to the European Nucleotide Archive under the project

accession PRJEB74486. *P. pacificus* strains and bacterial isolates generated in this work are freely available through the Lead Contact. Further information and requests for resources and reagents should be directed to and will be fulfilled by the lead contact, Ralf J. Sommer (ralf.sommer@tuebingen.mpg.de). Source data are provided with this paper.

## Code availability

All the code and data for Bayesian analysis are available at https://github.com/shielapearl18/Bayesian-estimates-of-mouth-form-responses-in-vitamin-B12-enriched-environment (https://doi.org/10.5281/zenodo.19002949).

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

## Acknowledgements

We would like to thank the entire support team and technicians of the Sommer lab in particular Heike Hausmann for assisting in freezing nematode cultures. We are also grateful to Drs. Adrian Streit and Catia Igreja for discussions and helpful comments on the manuscript. The work was funded by the Max Planck Society through institutional funds.

## Author contributions

Conceptualization, S.P.Q. and R.J.S.; Methodology, S.P.Q., A.K., C.R., and R.J.S; Formal analysis, S.P.Q., A.K.; Investigation, S.P.Q.; Resources, R.Z. and H.W.; Writing–original draft, S.P.Q. and R.J.S.; Writing–review & editing, S.P.Q., A.K., C.R., and R.J.S.; Visualization, S.P.Q. and A.K.; Funding acquisition, R.J.S; Supervision, R.J.S.

## Funding

## Competing interests

The authors declare no competing interests.
