## [Transparent Peer Review file · Nature Communications]

Vitamin B12 induces memory of predation through vitellogenin provisioning

Corresponding Author: Professor Ralf Sommer

Version 0:

Reviewer comments:

Reviewer #1

(Remarks to the Author)

Previous work from The Sommer lab identified vitamin B12 as an important factor in increasing predatory behavior in *P. pacificus*. Adding to that work, Quiobe et al., found that the function of vitamin B12 in the met/SAM cycle was needed for the transgenerational epigenetic inheritance (TEI) of the predatory mouth form. The authors identified upregulation of vitellogenin genes caused by a vitamin B12 rich diet and concluded that the change in nutritional provisioning to embryos was responsible for TEI of the predatory mouth form. This study uncovers an unexpected role of vitamin B12 in transgenerational memory of an animal's morphology and behavior.

Overall, the experimental design and results are clear and thorough. However, an alternative logical hypothesis has not been explored and there are several major and minor issues that need to be addressed to improve the paper.

Major comments:

Much of figure 1 presents experiments that were previously published and is discussed in the introduction. This is highly unusual, and we wonder whether it could just be presented in a cartoon, rather than re-showing data.

The first experiment of this paper shows that both female and male gametes can cause TEI of the predatory mouth form. There is no follow up from this experiment and the results don't contribute further to the paper. This experiment could be removed entirely, or they should discuss more about what this could mean in terms of vitamin B12 and/or methionine as a memory.

The authors use a very high dose of 1,000 nM vitamin B12 and state that this is equivalent to that used in our Watson et al Cell 2013 paper, and they say this is common in *C. elegans* studies. This is not correct, the maximum dose used in the Watson paper (and most of our other papers, e.g., see also Bulcha et al, for a titration) Cell was 64 nM. This is a critical point because a high dose of vitamin B12 is required to observe a transgenerational effect and reducing this dose decreases the number of generations that the predatory mouth form is observed. This therefore suggests that the 'memory' phenotype is caused simply by passing vitamin B12 from mother (or father) to offspring. If this is true, the use of the term "transgenerational epigenetic inheritance" is not correct. The authors could test this hypothesis by using an *mrp-5* mutant (assuming there is an ortholog in *P. pacificus*) because in *C. elegans* MRP-5 transports vitamin B12 from the mother to the offspring (Na et al, Cell Reports 2018).

The Walker lab has shown that SAMS-1 converts methionine into S-adenosylmethionine which is required to make phosphatidylcholine, an important component of membranes. The authors should examine whether this plays a role in the phenotypes they observe by testing a *sams-1* mutant.

How and why are vitellogenin genes upregulated by vitamin B12? Does methionine activate the same genes? And is this related to any (histone) methylation, which is the function of the Met/SAM cycle and an epigenetic mark, or only methionine?

What could be the ecological relevance of vitamin B12 activating the predatory form and having a transgenerational effect? And would vitamin B12 concentrations be sufficiently high in the wild?

Minor comments:

Did the authors control their gene expression data for developmental effects caused by vitamin B12? Also, the gene expression tables are difficult to navigate.

Replace “vitamin B9” with “folate” in the abstract (or both) as readers will be more familiar with that.

In figure 2 the authors describe some panels out of order. (e.g. figure 2e before figure 2c).

Figures are inconsistently indicated in the text (e.g. Fig. 2e vs. Figure 2E).

Authors did not mention if NGM concentration was adjusted to account for discrepancy in osmolarity and nutrient amount when adding supplements.

mce-1 is indicated at the wrong reaction in Figure 3.

Page 11 – it would be helpful to mention what vitellogenins are in this section.

Page 13 – vitamin B12 can also be made by certain archaea, not only by bacteria.

Gabby Giese & Marian Walhout

Reviewer #2

(Remarks to the Author)

Quiobe et al. investigate the mechanisms of the transgenerational epigenetic inheritance (TEI) of the memory of ancestral environment. To this end, they use as a model system the mouth dimorphism of the *Pristionchus pacificus* nematode, which they previously found to be modulated by diet (the presence of a *Novosphingobium* bacterium) in a transgenerational manner. Through a series of well-conceived and well-performed experiments the authors demonstrated that bacteria-derived vitamin B12 is both necessary and sufficient for transgenerational memory; found a dose response of B12 with increasing concentration required for a transgenerational inheritance of the predatory morph during an increasing number of generations. They also found that the TEI requires methionine and methionine-synthase *metr-1* but not folate. Finally, they found that vit B12 (likely) acts through increased provisioning of vitellogenin. Overall, the results are interesting and the conclusions are generally well supported.

However, I have several major and minor points to address before publication.

The main problem I have with this manuscript is that figure 1 which is almost entirely recycled from their recent paper (specifically panels b, c which are taken from Fig 1 c,d and fig 2 B-F of Quiobe et al 2025 Sci adv). While it is important to put the new finding in context with their previous findings, I think there is no need (and it is inappropriate) to include figures with data already published as it could mislead reader into thinking they are new results, unless clearly stated and an attribution to the original source next to the reproduced figure is clearly specified and properly referred to. Nevertheless, I think a scheme indicating the previous finding is enough to put the novel findings in context.

A second major point is how do the authors reconcile that both female and male gametes transmit TEI of the predatory mouth form with the proposed mechanism based on increased vitellogenin provisioning which would implicate only oocytes?

What do the authors make of the collagen expression changes with collagen being the most significantly overrepresented category among differentially expressed genes? It is usually a red flag for uncontrolled variation in developmental stage between transcriptomic samples.

Does increased vit transcription also translate into increased vit provisioning to the embryo?

Likely yes but it would be nice to check using protein tags or even by using autofluorescence (see Perez et al 2017 Nature)

Moreover, given that vitellogenins are constituent of a lipoprotein complex (similar to LDL) that carries lipids, the actual signal could be a lipid or even broad change in the composition of the lipids carried by the complex (and dependent on *rme-2*) rather than an increased in amount, as it has been shown already in at least two cases of intergenerational transmission in *C. elegans* (Wang et al 2023 Nature cell biology, and Wilhelm et al. 2025 biorxiv doi: <https://doi.org/10.1101/2025.06.03.657568>). Considering that vitamin b12 and methionine are major regulators of lipid metabolism in nematodes I think the authors should at least discuss this possibility in the discussion section.

Version 1:

Reviewer comments:

Reviewer #1

(Remarks to the Author)

The authors have done an excellent job revising the paper based on our and the other reviewer's comments. We only have a few remaining comments to further improve the paper:

Most importantly, the notion that direct transfer of vitamin B12 from mother to offspring can contribute to the transgenerational effects is not sufficiently disproved, especially since transgenerational effects correlate with supplemented B12 concentrations. The experiment with *mrp-5* mutants shows that there is a bit of a decrease in the transgenerational effect (not dramatic). It would be possible that there are multiple transporters that deposit vitamin B12 in the offspring as well. We suggest a short paragraph expanding on this in the Discussion.

The authors use only micromolar doses of folate (presumably because folate has low solubility in water). It is possible this is simply not enough folate to induce the EU mouth-form. The authors should add a sentence addressing this, and mention whether they have some evidence to suggest micromolar doses of folate is sufficient to be uptaken by *P. pacificus*.

The use of both TEI and LTEI abbreviations can be confusing. Maybe one can be spelled out throughout? (very minor)

Line 144: the sentence and symbol for the quantification are hard to understand, consider explaining/rephrasing?

How many animals were used for single animal RNA-seq? And how similar were the profiles among animals from the same experiment?

Line 184: change "12,2" to "12.2"

Line 234/235: methyl malonylCoA mutase is encoded by *mmcm-1*, not *mce-1*! Thus, the enzyme that uses vitamin B12 as a cofactor in propionate breakdown has not been directly studied here.

Figure 3C needs no vitamin B12 in the plot.

Line 254: change "...indicating the Ppa..." to "...indicating that Ppa..."

We suggest a three-way Venn diagram to convey the differentially expressed genes for diet, vitamin b12, and methionine.

Line 344/345 "...harvesting worm plates with preferentially adult individuals." – rephrase, worms were harvested not plates and preferentially adult is weird.

Gabby Giese & Marian Walhout

Reviewer #2

(Remarks to the Author)

The authors have addressed my main concerns and I am happy with the current version of the manuscript

Version 2:

Reviewer comments:

Reviewer #1

(Remarks to the Author)

We are happy with the revision.

Point-by-point response:

Reviewer #1 (Remarks to the Author):

Previous work from The Sommer lab identified vitamin B12 as an important factor in increasing predatory behavior in *P. pacificus*. Adding to that work, Quiobe et al., found that the function of vitamin B12 in the met/SAM cycle was needed for the transgenerational epigenetic inheritance (TEI) of the predatory mouth form. The authors identified upregulation of vitellogenin genes caused by a vitamin B12 rich diet and concluded that the change in nutritional provisioning to embryos was responsible for TEI of the predatory mouth form. This study uncovers an unexpected role of vitamin B12 in transgenerational memory of an animal's morphology and behavior.

Overall, the experimental design and results are clear and thorough. However, an alternative logical hypothesis has not been explored and there are several major and minor issues that need to be addressed to improve the paper.

Response: We thank the reviewer for the overall judgement of our work.

Major comments:

Much of figure 1 presents experiments that were previously published and is discussed in the introduction. This is highly unusual, and we wonder whether it could just be presented in a cartoon, rather than re-showing data.

Response: We have followed this recommendation and have replaced Figure 1 with a cartoon. The reason for having Figure 1 in the first place was that the experimental design used in our study system is relatively novel and different from that of other studies. We just wanted to make sure that readers are not lost early on. But given that also reviewer 2 had a similar concern, we are happy to change figure 1 into a cartoon like summary.

The first experiment of this paper shows that both female and male gametes can cause TEI of the predatory mouth form. There is no follow up from this experiment and the results don't contribute further to the paper. This experiment could be removed entirely, or they should discuss more about what this could mean in terms of vitamin B12 and/or methionine as a memory.

Response: We thank the reviewer for this suggestion. We had similar concerns that this piece of data is unconnected to the rest of the story. Therefore, we have removed this paragraph all together.

The authors use a very high dose of 1,000 nM vitamin B12 and state that this is equivalent to that used in our Watson et al Cell 2013 paper, and they say this is common in *C. elegans* studies. This is not correct, the maximum dose used in the Watson paper (and most of our other papers, e.g., see also Bulcha et al, for a titration) Cell was 64 nM. This is a critical point because a high dose of vitamin B12 is required to observe a transgenerational effect and reducing this dose decreases the number of generations that the predatory mouth form is observed. This therefore suggests that the 'memory' phenotype is caused simply by passing vitamin B12 from mother (or father) to offspring. If this is true, the use of the term "transgenerational epigenetic inheritance" is not correct. The authors could test this hypothesis by using an *mrp-5* mutant (assuming there is an ortholog in *P. pacificus*) because in *C. elegans* MRP-5 transports vitamin B12 from the mother to the offspring (Na et al, Cell Reports 2018).

Response: We thank the reviewer for these comments. We have now clarified our wording and have addressed all these issues:

- 1.) We had mistaken the reference of our work (Akduman, ref #15) with the *C. elegans* work by Watson et al (reference #17) when introducing the high vitamin B12 concentrations. We have now corrected this by saying that we started with vitamin B12 concentrations similar to previous *Pristionchus* work. Specifically, work in our lab had used 500 nM in the context of killing efficiency (Akduman et al, 2020). When starting to work with vitamin B12 in the context of mouth-form plasticity, we started to also increase the concentration to 1,000 nM and 1,500 nM vitamin B12 and observed the strongest and most consistent effect with these concentrations. We have now clarified these issues in the text.
- 2.) Lower concentrations, such as those used in *C. elegans*, cause strong induction but no transgenerational inheritance, indicating that induction and memory require different concentrations of vitamin B12. This is now properly described on pages 5/6 and 9 and in Figures 2 & 4.
- 3.) The effect with 1,000 nM and 1,500 nM vitamin B12 are still a transgenerational inheritance as it lasts for three generations after stopping the supplementation with vitamin B12. In line with this, 500 nM gives intergenerational inheritance, whereas smaller concentrations have no effect. In our discussion, we point out – along the same lines as the reviewer – that these are very high doses of vitamin B12 and we discuss various boundary conditions.
- 4.) We added two more concentrations of vitamin B12 in our supplementation experiments. At 0.1 nM vitamin B12, we still see induction of the predatory morph. In contrast, 0.01 nM vitamin B12 does not cause any response. We have now added these data to our analysis.

5.) We had indeed generated and used *mrp-5* mutants in *P. pacificus*. However, this mutant shows a wild type pattern of transgenerational inheritance (now presented as Fig. 5a,b), indicating that factors other than the direct passing of vitamin B12 from parents to offspring are ultimately responsible for the observed memory. We now introduce these findings in Figure 5 and on pages 11 and 12.

The Walker lab has shown that SAMS-1 converts methionine into S-adenosylmethionine which is required to make phosphatidylcholine, an important component of membranes. The authors should examine whether this plays a role in the phenotypes they observe by testing a *sams-1* mutant.

Response: This is a great suggestion that also came to our mind. Indeed, we had planned to use mutations in the *sams-1* gene. Unfortunately, *P. pacificus* has a massive expansion of the *sams-1* gene with additional copy number variations between strains. Therefore, we cannot target this gene in CRISPR experiments.

How and why are vitellogenin genes upregulated by vitamin B12? Does methionine activate the same genes? And is this related to any (histone) methylation, which is the function of the Met/SAM cycle and an epigenetic mark, or only methionine?

Response: We thank the reviewer for this suggestion. We had not looked at gene expression after methionine supplementation. We have now added these experiments and have – also in response to reviewer 2 – moved for some experiments from mixed stage RNAseq to single worm RNAseq of day 1 adult animals. This protocol had been previously established in our lab. As a result, parts of Figure 2 have been changed accordingly, and the methionine data have been added to Fig. 4. This also addresses a major concern of reviewer 2.

What could be the ecological relevance of vitamin B12 activating the predatory form and having a transgenerational effect? And would vitamin B12 concentrations be sufficiently high in the wild?

Response: We thank the reviewer for bringing up this question. The induction of the predatory form by vitamin B12 is likely of importance in the competitive environment of the decaying beetle ecosystems in which *Pristionchus* nematodes are found. There are multiple competing nematodes on beetle carcasses and *Pristionchus* has an advantage through intraguild predation. Also, previous work from our lab has shown that *Pristionchus* selectively feeds on vitamin B12 producing bacteria when exposed to a mixture of diverse bacteria (Lo et al., Nature Communications, 2023), a striking finding that is currently further studied by my previous postdocs. We have now added a novel paragraph to the discussion section to address these issues.

Minor comments:

Did the authors control their gene expression data for developmental effects caused by vitamin B12? Also, the gene expression tables are difficult to navigate.

Response: As indicated above, we have redone gene expression analysis with single worm RNAseq. Also, we changed the representation of the data.

Replace “vitamin B9” with “folate” in the abstract (or both) as readers will be more familiar with that.

Response: Done.

In figure 2 the authors describe some panels out of order. (e.g. figure 2e before figure 2c).

Response: Corrected.

Figures are inconsistently indicated in the text (e.g. Fig. 2e vs. Figure 2E).

Response: Corrected.

Authors did not mention if NGM concentration was adjusted to account for discrepancy in osmolarity and nutrient amount when adding supplements.

Response: Corrected.

mce-1 is indicated at the wrong reaction in Figure 3.

Response: Corrected.

Page 11 – it would be helpful to mention what vitellogenins are in this section.

Response: We have added an explanatory sentence.

Page 13 – vitamin B12 can also be made by certain archaea, not only by bacteria.

Response: Corrected.

Gabby Giese & Marian Walhout

Reviewer #2 (Remarks to the Author):

Quiobe et al. investigate the mechanisms of the transgenerational epigenetic inheritance (TEI) of the memory of ancestral environment. To this end, they use as a model system the mouth dimorphism of the *Pristionchus pacificus* nematode, which they previously found to be modulated by diet (the presence of a *Novosphingobium* bacterium) in a transgenerational manner. Through a series of well-conceived and well-performed experiments the authors demonstrated that bacteria-derived vitamin B12 is both necessary and sufficient for transgenerational memory; found a dose response of B12 with increasing concentration required for a transgenerational inheritance of the predatory morph during an increasing number of generations. They also found that the TEI requires methionine and methionine-synthase *metr-1* but not folate. Finally, they found that vit B12 (likely) acts through increased provisioning of vitellogenin. Overall, the results are interesting and the conclusions are generally well supported.

Response: We thank the reviewer for the overall judgement of our work.

However, I have several major and minor points to address before publication.

The main problem I have with this manuscript is that figure 1 which is almost entirely recycled from their recent paper (specifically panels b, c which are taken from Fig 1 c,d and fig 2 B-F of Quiobe et al 2025 Sci adv). While it is important to put the new finding in context with their previous findings, I think there is no need (and it is inappropriate) to include figures with data already published as it could mislead reader into thinking they are new results, unless clearly stated and an attribution to the original source next to the reproduced figure is clearly specified and properly referred to. Nevertheless, I think a scheme indicating the previous finding is enough to put the novel findings in context.

Response: We apologize if the reviewer got the wrong impression. It was never our intention to mislead the reader. As indicated above, we just wanted to make sure that readers understand our novel and complex study design.

We have now replaced Figure 1 by a graphical representation summarizing our previous findings.

A second major point is how do the authors reconcile that both female and male gametes transmit TEI of the predatory mouth form with the proposed mechanism based on increased vitellogenin provisioning which would implicate only oocytes?

Response: As suggested by reviewer 1, we have removed this entire paragraph, also because it was not connected to the rest of our study.

What do the authors make of the collagen expression changes with collagen being the most significantly overrepresented category among differentially expressed genes? It is usually a red flag for uncontrolled variation in developmental stage between transcriptomic samples.

Response: We thank the reviewer for bringing up this question. Previous work from our lab has shown that the change in mouth form (Eu vs ST) results in substantial gene expression changes, including several collagens (Sieriebriennikov et al., *PLoS Genetics*, 2020, Sun et al., *Genome Research*, 2022; Theska et al., *Evolution&Development*, 2024). We are currently working on some of these genes as interesting markers and protein constituents of the teeth.

As already indicated in response to reviewer 1, we have redone this part of the differential gene expression analysis and used single worm transcriptomics rather than mixed stage cultures. As indicated above, parts of Figure 2 have been replaced, and additional analysis has been added to Figure 4.

Does increased vit transcription also translate into increased vit provisioning to the embryo?
Likely yes but it would be nice to check using protein tags or even by using autofluorescence (see Perez et al 2017 *Nature*)

Response: Unfortunately, autofluorescence does not work in *P. pacificus* due to differences in lipid composition.

We had aimed for quite some time in tagging individual *vit* genes, which turned out to be inheritantly more difficult in *P. pacificus* than *C. elegans*, likely due to the nine independent copies of *vit-6*, that are all very similar to each other in sequence. We finally managed to obtain one line with a single (rather than two) ALFA tag at the C-terminus of the protein. Unfortunately, it has a phenotype on its own. While we keep trying to generate a flagged protein, for the time being this experiment is unfeasible in *P. pacificus*.

Moreover, given that vitellogenins are constituent of a lipoprotein complex (similar to LDL) that carries lipids, the actual signal could be a lipid or even broad change in the composition of the lipids carried by the complex (and dependent on *rme-2*) rather than an increased in amount, as it has been shown already in at least two cases of intergenerational transmission in *C. elegans* (Wang et al 2023 Nature cell biology, and Wilhelm et al. 2025 biorxiv doi: <https://doi.org/10.1101/2025.06.03.657568>). Considering that vitamin b12 and methionine are major regulators of lipid metabolism in nematodes I think the authors should at least discuss this possibility in the discussion section.

Response: We thank the reviewer for bringing up this point and we have added this to our Discussion.

In summary, we feel that we have been able to successfully respond to the majority of concerns (except for those that were technically not feasible. We feel that this has increased the quality of the data. Therefore, we want to thank the reviewers once again for their input.

Thank you very much for consiedring this revised version.

Ralf J. Sommer, on behalf of all co-authors

Reviewer #1 (Remarks to the Author):

The authors have done an excellent job revising the paper based on our and the other reviewer's comments. We only have a few remaining comments to further improve the paper:

Response: We thank the reviewer for this overall statement.

Most importantly, the notion that direct transfer of vitamin B12 from mother to offspring can contribute to the transgenerational effects is not sufficiently disproved, especially since transgenerational effects correlate with supplemented B12 concentrations. The experiment with *mrp-5* mutants shows that there is a bit of a decrease in the transgenerational effect (not dramatic). It would be possible that there are multiple transporters that deposit vitamin B12 in the offspring as well. We suggest a short paragraph expanding on this in the Discussion.

Response: We have added a paragraph to the discussion as suggested (page 15), writing the following:

The experiments with *Ppa-mrp-5* mutants show a normal TEI suggesting that it is not the direct transmission of vitamin B12 but a downstream factor causing the observed memory. However, the transgenerational effect is slightly decreased compared to wild type and one possibility that cannot be completely ruled out is that multiple transporters in addition to *Ppa-mrp-5* are involved in vitamin B12 transmission. Nonetheless, *Ppa-mrp-5* is one-to-one orthologous to *Cel-mrp-5*; thus, there are no additional copies of this transporter in the *P. pacificus* genome.

The authors use only micromolar doses of folate (presumably because folate has low solubility in water). It is possible this is simply not enough folate to induce the EU mouth-form. The authors should add a sentence addressing this, and mention whether they have some evidence to suggest micromolar doses of folate is sufficient to be uptaken by *P. pacificus*.

Response: We thank the reviewers for bringing up this point and have now rephrased as follows:

First, we used 10 μM , 50 μM and 100 μM of folate (vitamin B9) for supplementation of standard *E. coli* plates using similar concentrations as a previous study in *C. elegans*.⁴¹ However, we found no credible induction of the predatory mouth form (e.g., for 100 μM , $0.384 \leq \text{HDI}(\theta_{F2}) \leq 0.755$, Fig. 4d-f). Note that this might be due to the micromolar doses of folate that had to be used in these experiments due to solubility.

The use of both TEI and LTEI abbreviations can be confusing. Maybe one can be spelled out throughout? (very minor)

Response: We have spelled out long-term environmental induction experiment throughout the manuscript.

Line 144: the sentence and symbol for the quantification are hard to understand, consider explaining/rephrasing?

Response: We have changed this sentence accordingly, writing the following:

Specifically, in three independent biological replicates with a total of 60 assay plates we found high but incomplete induction in the first generation after vitamin B12 supplementation ($0.878 \leq \text{HDI}(\theta) \leq 0.994$, where θ is the inferred probability of developing the predatory mouth form).

How many animals were used for single animal RNA-seq? And how similar were the profiles among animals from the same experiment?

Response: We used five replicates for all single animal RNA-seq experiments. The results were highly overlapping as already seen in the original paper where we established this method by using three replicates. This information is provided in the Materials and Methods section.

Line 184: change "12,2" to "12.2"

Changed.

Line 234/235: methyl malonylcoA mutase is encoded by mmcm-1, not mce-1! Thus, the enzyme that uses vitamin B12 as a cofactor in propionate breakdown has not been directly studied here.

We are sorry for this confusion and have now changed accordingly:

In *P. pacificus* as in *C. elegans*, the methionine-synthase is encoded by *Ppa-metr-1*. To manipulate the methylmalonyl coenzyme A (CoA) mutase pathway, we targeted the upstream factor *Ppa-mce-1*, that is one-to-one orthologous to *Cel-mce-1*. We targeted *Ppa-metr-1* and *Ppa-mce-1* in *P. pacificus* RSC011 by CRISPR to study a potential role in dietary induction and TEI of the predatory mouth form (Fig. 3b).

Figure 3C needs no vitamin B12 in the plot.

Response: Thanks for pointing this confusion out. Figure 3C shows the response of wild type and mutant animals to different concentrations of vitamin B12 supplementation. This is the only graph that has concentrations on the X-axis, rather than generations. We changed the outline to make that clear.

Line 254: change "...indicating the Ppa..." to "...indicating that Ppa..."

Changed accordingly.

We suggest a three-way Venn diagram to convey the differentially expressed genes for diet, vitamin b12, and methionine.

Response: We changed to a three-way Venn diagram.

Line 344/345 "...harvesting worm plates with preferentially adult individuals." – rephrase, worms were harvested not plates and preferentially adult is weird.

Changed accordingly:

.....by harvesting cultures with preferentially adult individuals.

Gabby Giese & Marian Walhout

Reviewer #2 (Remarks to the Author):

The authors have addressed my main concerns and I am happy with the current version of the manuscript

Response: We thank the reviewer for this overall statement.